# Incentive preferences for community health volunteers in Kenya: findings from a discrete choice experiment

Timothy Abuya ![ORCID],[1] Daniel Mwanga,[1] Melvin Obadha,[2,3] Charity Ndwiga,[1] George Odwe ![ORCID],[1] Daniel Kavoo,[4] John Wanyugu,[4] Charlotte Warren,[5] Smisha Agarwal[6]

[1]Reproductive Health, Population Council, Nairobi, Kenya
[2]Health Economics Research Centre, Nuffield Department of Population Health, University of Oxford, Oxford, UK
[3]Health Economics Research Unit, KEMRI-Wellcome Trust Research Programme, Nairobi, Kenya
[4]Division of Community Health Services, Minsitry of Health, Nairobi, Kenya
[5]Population Council, Washington, DC, USA
[6]Department of International Health, Johns Hopkins Bloomberg School of Public Health, Baltimore, Maryland, USA

**Correspondence to**
Dr Timothy Abuya;
tabuya@popcouncil.org

## ABSTRACT

**Background** Community health volunteers (CHVs) play crucial roles in enabling access to healthcare at the community levels. Although CHVs are considered volunteers, programmes provide financial and non-financial incentives. However, there is limited evidence on which bundle of financial and non-financial incentives are most effective for their improved performance.

**Methods** We used a discrete choice experiment (DCE) to understand incentive preferences of CHVs with the aim to improve their motivation, performance and retention. Relevant incentive attributes were identified through qualitative interviews with CHVs and with their supervisors. We then deployed a nominal group technique to generate and rank preferred attributes among CHVs. We developed a DCE based on the five attributes and administered it to 211 CHVs in Kilifi and Bungoma counties in Kenya. We used mixed multinomial logit models to estimate the utility of each incentive attribute and calculated the trade-offs the CHWs were willing to make for a change in stipend.

**Results** Transport was considered the incentive attribute with most relative importance followed by tools of trade then monthly stipend. CHVs preferred job incentives that offered higher monthly stipends even though it was not the most important. They had negative preference for job incentives that provided award mechanisms for the best performing CHVs as compared with jobs that provided recognition at the community level and preferred job incentives that provided more tools of trade compared with those that provided limited tools.

**Conclusion** A bundled incentive of both financial and non-financial packages is necessary to provide a conducive working environment for CHVs. The menu of options relevant for CHVs in Kenya include transport, tools of trade and monthly stipend. Policy decisions should be contextualised to include these attributes to facilitate CHW satisfaction and performance.

## INTRODUCTION

The shortage of health workers is worsening globally estimated at 18 million health workers by 2030 and 4.3 million in Africa and Asia.[1–3] Recruitment and training of community health workers (CHWs) increase access to basic health services including health education where the formal sector

### Strengths and limitations of this study

► Discrete choice experiment (DCE) enabled us to study incentive packages that did not exist and quantify community health volunteers (CHVs) preferences for the attributes of incentive alternatives, relative importance they placed on the attributes and trade-offs of incentive attributes CHVs were willing to make.

► Data generated from this method provide useful information for policymakers on the attributes of a good incentive package for CHVs in Kenya and other similar contexts.

► DCE results illustrated heterogeneity, illustrating the value of diversity and how that can be used to ensure context-specific adaptation of the menu of incentives that a country can adopt to make use of limited resources.

► DCE methods are prone to bias as they present hypothetical alternatives to respondents. Furthermore, our DCE adopted a forced-choice elicitation format (lack of opt-out) which might have exacerbated hypothetical bias and affected our willingness to accept estimates.

falls short.[1 4 5] Although CHWs roles, recruitment, remuneration and training vary,[1] they work within their own community to promote health.[6] CHW programmes have reported substantial improvements in healthcare outcomes but struggled to maintain quality at scale.[5 7 8] While CHW are considered volunteers,[9] programmes provide financial and non-financial incentives,[10] facilitating their retention and improved performance.[11–13] Integrating CHW into the formal health sector requires adequate incentives.[14–16] However, limited evidence exist on which bundle of financial and non-financial incentives are most effective.[5 17]

In Kenya, community health services (CHS) are implemented through community units (CUs), each serving a population of 5000 people. Community health volunteers (CHV)

who serve these units are chosen by the community and trained to create demand for preventive services. CHV are supervised by a community health extension workers (CHEW) who provide training and technical support. Currently Kenya has an estimated 6359 CUs and 63 590 CHVs. There are 1500 CHEWs.[18] CHS in Kenya increase attendance of antenatal care visits, deliveries by skilled birth attendants, intermittent preventive treatment, testing for HIV during pregnancy, exclusive breastfeeding and hygiene practices.[19 20] Despite this evidence of effectiveness, CHVs are considered as volunteer health workers. The government directed that they are paid a stipend of KES 2000/month (~US$20); however, this compensation continues to be ad hoc mostly from non-governmental organisations (NGO) and development partners. This is often in the form of transport reimbursement, lunches or a monthly stipend that is fixed or performance based. There are efforts to mainstream remuneration of CHVs with a few counties legislating county community health bills that sets aside funding for CHS.

Stated preference elicitation methods, such as discrete choice experiments (DCE) can help better understanding of incentive preferences and trade-offs CHVs are willing to make.[21] Financial remuneration as well as non-financial incentives are critical to improve performance and retention of CHVs, as recommended by the 2018 WHO guideline on health policy and system support to optimise CHW programmes.[17] This study uses a DCE to understand incentive preferences of CHVs with the aim to improve motivation, performance, and retention of CHVs. Although DCE methods have been applied in health research in high-income countries, there are few examples of DCE used in low-income setting for various cadres of health workers[22–27] and especially among CHWs.[5 28 29] This study provides empirical evidence on the job incentive structure necessary to contribute to the welfare of CHVs using DCE.

## METHODS

### Phase I: identifying attributes and levels

The study was conducted among CHVs from two subcounties of Kilifi, a rural coastal county, and Bungoma, a western county. Both counties reflect a range of commonly observed barriers to care, including geographical access constraints and cultural vulnerabilities.[30] The counties were purposefully selected in collaboration with County Health Management Teams (CHMTs) to include a functional CHS where CHVs conduct routine visits, collect data and receive some form of incentives. Additionally, the selected sites had several collaborative projects between NGOs and the government being implemented to strengthen local CHS through capacity building and improve access to health information.

Phase I identified financial and non-financial attributes and levels that influence CHV performance. We conducted four focus group discussions (FGDs) with CHVs and another four with their supervisors, the CHEWs. FGDs examined their understanding of incentives, preferred incentives and barriers to implementation and the feasibility of implementing the incentives. We then deployed a nominal group technique to generate and rank preferred attributes among CHVs and CHEWs. At the national level, we conducted four in-depth interviews with policymakers at the division of community health and three key organisations that implement CHS programmes.

### Validation and refining attributes

Seventeen attributes and their corresponding levels were generated and subjected to a validation process by partners and policymakers at the division of community health. Key considerations used were relevance, ability to compute willingness to accept (WTA) measures, correlations between attributes, importance of an attribute, plausibility, capability of being traded and attribute non-attendance.[31 32] The process resulted in eight attributes, which were further reduced to five through a pilot study (table 1).

### Experimental design and construction of choice tasks

Phase II comprised a cross-sectional quantitative DCE survey among CHVs to elicit their preferences. The choice experiment was unlabeled consisting of two hypothetical incentive alternatives in a forced-choice elicitation format, which included full profiles where all five attributes appeared in each alternative. A fractional factorial experimental design was used to generate 12 choice tasks. An orthogonal design was used where each attribute was statistically independent of each other and balanced using Sawtooth Software.[33] A check for dominant alternatives was conducted and included as an extra choice task to act as a rationality test and assess internal validity of the data.[34] The choice tasks were designed into a survey questionnaire. Figure 1 shows a sample of the choice set that was presented to the participants.

### DCE survey questionnaire

The survey questionnaire covered background characteristics, sources of income, current incentive structure and compensation, workload, supervision, training and the choice tasks. The research assistants were trained on how to explain each attribute and levels to the participants. They introduced the choice tasks by explaining to the participant the different scenarios and asked, given a choice, which of the alternatives they would prefer. This was repeated for each of the choice tasks with participants selecting preferred alternative. The choice tasks were programmed to appear in a random order to minimise social desirability bias.[34]

### Sampling

A rule of thumb by Johnson and Orme[35] was used to determine the minimum sample size represented in the equation below:

$$N > \frac{500c}{(t* \ a)}$$

**Table 1** Attributes and attribute levels included in the DCE

| Attribute | Description | Attribute level | Coding |
|---|---|---|---|
| Recognition | Defined as any form of mechanism that help CHV be recognised at either community or facility level. This was described either as an award system or given priority in various community meetings | Recognition at community level (recognition during public meetings) | 0 (ref.) |
| | | Recognition at facility (priority for service provision for them and their families, opportunity to work at facility as volunteer, and acknowledgement of CHV referrals and feedback) | 1 |
| | | Award mechanism for the best performing CHVs (certification, provision of gifts, wall of fame and opportunities for exchange visit) | 2 |
| Income-generating activities | Some form of seed money to support income-generating activities that are locally relevant | Provision of seed money/Grant KES 100 000/CU | 0 (ref.) |
| | | Provision of seed money/Grant KES 150 000/CU | 1 |
| Monthly stipend | Financial renumeration that is payable on monthly basis | KES 2500* | Continuous |
| | | KES 4000 | |
| | | KES 5500 | |
| Transport | Any form of support that will facilitate CHV movement from one place to another. It could be bicycles or motorcycles | Bicycles for CHV | 0 (ref.) |
| | | Motorcycles for CHV/CU | 1 |
| | | Motorcycle for community health extension workers and bicycle for CHV | 2 |
| Tools of trade | Availability of supplies ranging from commodities for promotive preventive activities for example, drugs, job aids or IEC materials, items that identify CHV in communities or safety gears such as raincoats to support during bad weather | Supplies and commodities+non-pharmaceutical+job aids/IEC materials | 0 (ref.) |
| | | Supplies and commodities+non-pharmaceutical+job aids/IEC materials+identification (badges and branded jacket/bag) | **1** |
| | | Supplies and commodities+non-pharmaceutical+job aids/IEC materials+identification (badges and branded jacket/bags)+safety gears (raincoats, gumboots and umbrella) | **2** |

*US$1=KES 100.
CHV, community health volunteer; CU, community unit; DCE, discrete choice experiment; IEC, information, education and communication.

Using a main effects model, a minimum sample size $N$ of 84 was derived where $c$ (largest number of levels $c$ among the five attributes) was 4, $t$ was 12 choice tasks and $a$ was 2 alternatives. To derive the sample size, CHVs were sampled from two socioeconomically and culturally diverse counties namely Kilifi and Bungoma. From each county, two subcounties were selected in consultation with the respective CHMTs. In Kilifi, the study was conducted in Kilifi North and Kaloleni subcounties, while in Bungoma, the study was conducted in Tongaren and Webuye West. Potential participants were recruited from all active CHWs selected with the help of the local community focal person. From each ward, a maximum of 13–14 active CHVs were targeted for the survey. From this process, 211 CHVs were identified (109 in Kilifi and 102 in Bungoma). We administered the DCE to all CHVs after obtaining consent. All who turned up agreed to complete the DCE survey, yielding 100% response rate.

### Data collection

Data collection was done electronically using the Open Data Kit programme by trained interviewers between 14 October and 22 October 2019. Before data collection, a field pretest was conducted to practice the tool and gather experiences on the best approaches of how to explain the choice tasks. The experiences from the field pretest were used by the data collection team to refine how to explain the choice tasks to the CHVs. CHVs were presented with 12 choice tasks prompting them to choose an option they preferred among two incentive alternatives.

### Data analysis

Analysis of DCE data follows random utility theory described in the published study protocol.[5] A mixed multinomial logit model (MMNL) was used to estimate choice probabilities and was preferred over a conditional logit model because it fully relaxes the assumption of independence of irrelevant alternatives (IIA), considers inter respondent preference heterogeneity and within

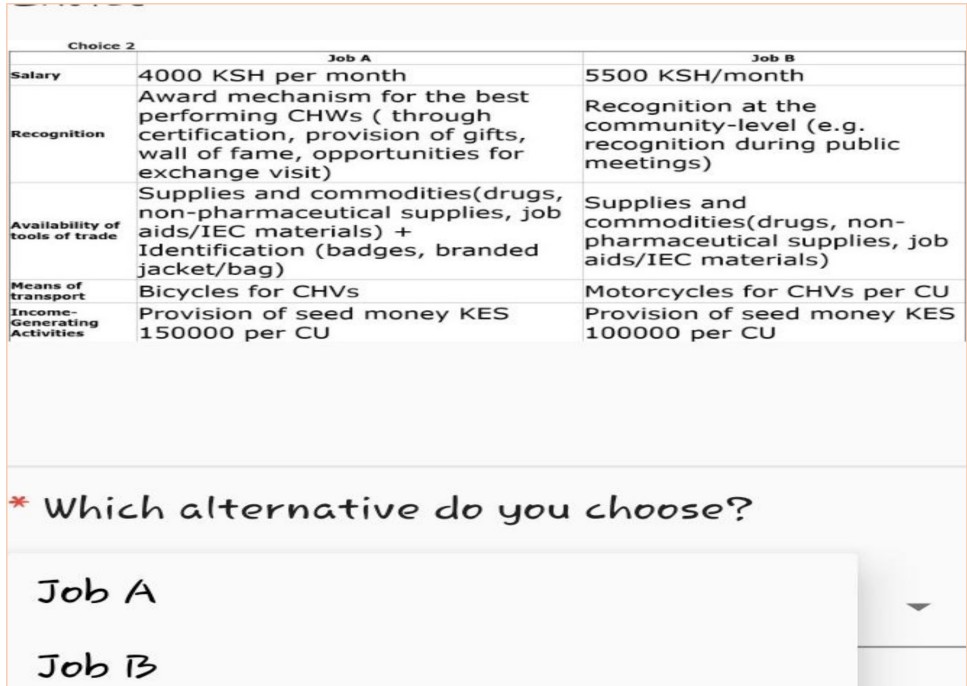

**Figure 1** Sample of choice set. CU, community unit; CHV, community health volunteer; CHW, community health worker; IEC, information, education and communication.

respondent correlation.[33][36] The utility function of the MMNL main effects model was specified as follows:

$$
\begin{aligned}
U_{njt} = \quad & \beta_0 + \beta_1 * Recognition\ at\ facility_{njt} + \beta_2 * Award\ mechanism_{njt} + \beta_3 \\
& * Income\ Generating\ Activities_{njt} + \beta_4 * Monthly\ stipend_{njt} + \beta_5 \\
& * Motorcycle_{njt} + \beta_6 * Motorcycle\ and\ Bicycle_{njt} + \beta_7 * Identification_{njt} \\
& + \beta_8 * Identification\ and\ safety\ gear_{njt} + \varepsilon_{njt}
\end{aligned}
$$

Where $U_{njt}$ was the utility a CHV $n$ derived from selecting incentive alternative $j$ in choice task t, $\beta_0$ was the alternative specific constant for alternative A, $\beta_1 - \beta_8$ were parameters to be estimated, $\varepsilon_{njt}$ were error terms, which were assumed to be IID following type 1 extreme value distribution, monthly stipend was a continuous attribute in KES, the rest were dummy coded variables for levels of the recognition, transport, income-generating activities (IGA) and tools of trade attributes. In the model, all parameters of the attributes were treated as random and normally distributed except monthly stipend, which was restricted to a lognormal distribution. The MMNL model resulted in means which represented choice probabilities or preferences and SD, which captured preference heterogeneity. The models were stratified by the two counties. Relative importance estimates were computed using the means from the panel MMNL main effects model. We took the absolute value of the mean of each attribute's parameters and multiplied it by the difference between the attribute levels' highest and lowest values.[37] This gave the maximum effect. The relative importance values were then calculated by considering the proportion of the maximum effect in the context of the total for each attribute.[37]

To further explore preference heterogeneity, we explored interaction effects in the panel MMNL model by introducing interactions between incentive attributes and CHVs' characteristics; monthly stipend and sex of CHV, monthly stipend and age of CHV, IGA and sex of CHV, and IGA and age of CHV. We computed WTA estimates using a panel MMNL model in willingness to pay (WTP) space.[38] We expressed the utility function in WTP space as follows

$$
\begin{aligned}
U_{njt} = \quad & \beta_{4n}(Monthly\ stipend_{njt} + \beta_0 + \beta_1 * Recognition\ at\ facility_{njt} + \beta_2 \\
& * Award\ mechanism_{njt} + \beta_3 * IGA_{njt} + \beta_5 \\
& * Motorcycle_{njt} + \beta_6 * Motorcycle\ and\ Bicycle_{njt} + \beta_7 \\
& * Identification_{njt} + \beta_8 * Identification\ and\ safety\ gear_{njt}) + \varepsilon_{njt}
\end{aligned}
$$

The monetary attribute was monthly stipend and represented the amount of money CHVs were willing to accept every month in KES. WTA estimates were stratified by county. All models used 1000 Halton draws. Analysis was conducted using Mixlogit and Mixlogitwtp commands on Stata V.15.1.[39] The $\chi^2$ test was used to compare the categories and t-test was used for continuous variables where appropriate.

## Patient and public involvement
There was no patient involvement.

## RESULTS
### Characteristics of respondents
Of 211 CHVs interviewed, the majority were women (70%), mean age was 46 years, and had secondary education (62%). More than a half (56%) reported aspiring to do business and about one-third wanted to continue working as

**Table 2** Characteristics of CHVs Interviewed

| | Kilifi | | Bungoma | | Total | | |
|---|---|---|---|---|---|---|---|
| | n=109 | % | n=102 | % | n=211 | % | P value |
| **Gender** | | | | | | | |
| Male | 38 | 34.9 | 26 | 25.5 | 64 | 30.3 | 0.139 |
| Female | 71 | 65.1 | 76 | 74.5 | 147 | 69.7 | |
| **Age (years) (mean (SD))*** | 45.5 (10.7) | | 46.5 (10.0) | | 46.0 (10.3) | | 0.458 |
| **Highest education level** | | | | | | | |
| Primary | 52 | 47.7 | 9 | 8.8 | 61 | 28.9 | <0.001 |
| Secondary | 47 | 43.1 | 84 | 82.4 | 131 | 62.1 | |
| College/university/vocational | 10 | 9.2 | 9 | 8.8 | 19 | 9 | |
| **Future career plans** | | | | | | | |
| Starting a business | 68 | 62.4 | 49 | 48 | 117 | 55.5 | 0.036 |
| Continue working as CHV | 47 | 43.1 | 21 | 20.6 | 68 | 32.2 | <0.001 |
| Other options | 25 | 22.9 | 28 | 27.5 | 53 | 25.1 | 0.45 |
| Health-related work | 17 | 15.6 | 21 | 20.6 | 38 | 18 | 0.346 |
| Further study | 10 | 9.2 | 6 | 5.9 | 16 | 7.6 | 0.367 |
| Become a political leader | 2 | 1.8 | 1 | 1 | 3 | 1.4 | 0.6 |
| **Marital status** | | | | | | | |
| Married/living together | 81 | 74.3 | 93 | 91.2 | 174 | 82.5 | |
| Widowed | 12 | 11 | 6 | 5.9 | 18 | 8.5 | 0.007 |
| Never married | 6 | 5.5 | 2 | 2 | 8 | 3.8 | |
| Divorced/separated | 10 | 9.2 | 1 | 1 | 11 | 5.2 | |
| **Working before becoming CHV (%)** | 92 | 84.4 | 99 | 97.1 | 191 | 90.5 | 0.002 |
| **Activities engaged before becoming CHV** | | | | | | | |
| Business owner | 49 | 53.3 | 41 | 41.4 | 90 | 47.1 | <0.001 |
| Agriculture | 12 | 13 | 63 | 63.6 | 75 | 39.3 | |
| Other | 32 | 34.8 | 15 | 15.2 | 47 | 24.6 | |
| Community/religious leader | 5 | 5.4 | 4 | 4 | 9 | 4.7 | |
| Teaching | 2 | 2.2 | 5 | 5.1 | 7 | 3.7 | |
| Other health professional | 1 | 1.1 | 5 | 5.1 | 6 | 3.1 | |
| Village savings group | 1 | 1.1 | 0 | 0 | 1 | 0.5 | |
| **Time taken to farthest household** | | | | | | | 0.004 |
| <15 min | 11 | 10.1 | 4 | 3.9 | 15 | 7.1 | |
| 15 min to <30 min | 44 | 40.4 | 27 | 26.5 | 71 | 33.6 | |
| 30 min to <60 min | 38 | 34.9 | 52 | 51 | 90 | 42.7 | |
| 1 hour to <2 hours | 15 | 13.8 | 11 | 10.8 | 26 | 12.3 | |
| ≥2 hours | 1 | 0.9 | 8 | 7.8 | 9 | 4.3 | |

*t-test was used to compare the age groups.
CHV, community health volunteer.

CHVs. The majority (91%) reported having been engaged in other economic activities before becoming CHV, mainly doing small scale business and agriculture (table 2).

### Current incentive structure for CHVs in Kenya
Table 3 shows that nearly all the CHVs interviewed (98%) received some form of compensation, either financial or non-financial, which were often infrequent (44.2%).

Among those reporting receipt of financial compensation, less than 15% received amounts that were within the recommended range of KES 2000 (US$20)/month while about 40% received about KES 5000 (US$50)/month.

### CHV preferences and relative importance estimates
Table 4 provides the preference estimates of the incentives attributes. Overall, as well as independently in Kilifi

**Table 3** Type of incentives currently provided

| | Kilifi | | Bungoma | | Total | | |
|---|---|---|---|---|---|---|---|
| | n=109 | % | n=102 | % | n=211 | % | P value |
| **% receiving** | | | | | | | |
| Any form of compensation | 107 | 98.2 | 100 | 98 | 207 | 98.1 | 0.947 |
| **Type of compensation received** | | | | | | | |
| Financial | 71 | 66.4 | 64 | 64 | 135 | 65.2 | 0.565 |
| Non-financial | 1 | 0.9 | 0 | 0 | 1 | 0.5 | |
| Both | 35 | 32.7 | 36 | 36 | 71 | 34.3 | |
| **Frequency of receipt of compensation** | | | | | | | |
| Monthly | 23 | 21.7 | 11 | 11 | 34 | 16.5 | <0.001 |
| Quarterly | 0 | 0 | 79 | 79 | 79 | 38.3 | |
| Semiannual | 2 | 1.9 | 0 | 0 | 2 | 1 | |
| Ad hoc | 81 | 76.4 | 10 | 10 | 91 | 44.2 | |
| **Amount received/month** | | | | | | | |
| <KES 1000 | 86 | 81.1 | 3 | 3 | 89 | 43.2 | <0.001 |
| KES 1000–2499 | 12 | 11.3 | 17 | 17 | 29 | 14.1 | |
| KES 2500–4999 | 5 | 4.7 | 1 | 1 | 6 | 2.9 | |
| KES 5000 or more | 3 | 2.8 | 79 | 79 | 82 | 39.8 | |
| **Sources of financial compensation** | | | | | | | |
| Government | 9 | 8.5 | 91 | 91 | 100 | 48.5 | <0.001 |
| NGO | 99 | 93.4 | 19 | 19 | 118 | 57.3 | <0.001 |
| Community members | 0 | 0.0 | 0 | 0.0 | 0 | 0.0 | NA |
| Do not know | 2 | 1.9 | 0 | 0 | 2 | 1 | NA |
| **Type of non-financial compensation** | | | | | | | |
| Food, clothes/other material goods | 33 | 91.7 | 10 | 27.8 | 43 | 59.7 | <0.001 |
| Supplies for work | 7 | 19.4 | 23 | 63.9 | 30 | 41.7 | <0.001 |
| Additional training | 2 | 5.6 | 12 | 33.3 | 14 | 19.4 | 0.003 |
| Other forms | 3 | 8.3 | 2 | 5.6 | 5 | 6.9 | 0.643 |
| Health services | 0 | 0 | 1 | 2.8 | 1 | 1.4 | NA |
| **Sources of non-financial compensation** | | | | | | | |
| Government | 10 | 27.8 | 3 | 8.3 | 13 | 18.1 | 0.032 |
| NGO | 29 | 80.6 | 31 | 86.1 | 60 | 83.3 | 0.527 |
| Community members | 1 | 2.8 | 4 | 11.1 | 5 | 6.9 | 0.164 |

US$1=KES 100.
NGO, non-governmental organisation.

and Bungoma counties, CHVs had a negative preference for jobs that provided motorcycles/CHU compared with those that provided bicycles only. Additionally, CHVs significantly preferred jobs that provided more tools of trade to those that provided limited tools. Overall, there was significant heterogeneity in preference across respondents for most attributes as denoted by the statistically significant SD.

Relative importance estimates indicate that transport was considered the most important incentive attribute followed by tools of trade and monthly stipend. The least important incentive attribute was IGA. However, in Kilifi county, CHV considered tools of trade as the most important attribute followed by transport and monthly stipend. This was slightly different from Bungoma county where CHVs rated transport as the most important attribute followed by monthly stipend and then tools of trade.

**WTA estimates of CHVs**
Table 5 shows WTA values in thousands of KES. Considering WTA estimates, in the whole sample, CHVs were significantly willing to accept a KES 1060 (US$10.60) increase in monthly stipend for an incentive package that awarded the best performing CHV to one that provided

**Table 4**  Main effects panel mixed multinomial logit model model: preference estimates

| Attribute | Attribute level | Whole sample | | Kilifi | | Bungoma | |
|---|---|---|---|---|---|---|---|
| | | β | SE | β | SE | β | SE |
| ASC | ASC | Ref. (0) | | | | | |
| | ASC μ | 0.138 | 0.192 | −0.191 | 0.277 | 0.555 | 0.320 |
| | ASC σ | −0.089 | 0.115 | −0.064 | 0.151 | −0.238 | 0.941 |
| Recognition | Recognition at community level | Ref. (0) | | | | | |
| | Recognition at facility μ | −0.039 | 0.175 | 0.350 | 0.259 | −0.466 | 0.300 |
| | Recognition at facility σ | 1.415** | 0.247 | 1.521** | 0.350 | 1.464** | 0.456 |
| | Award mechanism for best performing CHV μ | −0.682** | 0.186 | −0.585* | 0.278 | −0.819** | 0.277 |
| | Award mechanism for best performing CHV σ | 1.489** | 0.211 | 1.628** | 0.299 | 1.532** | 0.367 |
| Transport | Bicycles for CHV | Ref. (0) | | | | | |
| | Motorcycle for CHV/CU μ | −2.430** | 0.253 | −2.309** | 0.348 | −2.728** | 0.422 |
| | Motorcycle for CHV/CHU σ | 2.098** | 0.290 | 2.351** | 0.371 | 1.851** | 0.507 |
| | Motorcycle for CHEW and bicycle for CHV μ | 0.285 | 0.299 | −0.506 | 0.423 | 1.337* | 0.556 |
| | Motorcycle for CHEW and bicycle for CHV σ | 0.433 | 0.394 | 0.164 | 0.373 | −0.891 | 1.035 |
| Tools of trade | Supplies and commodities+non-pharmaceutical+job aids/IEC materials | Ref. (0) | | | | | |
| | Supplies and commodities+non-pharmaceutical+job aids/IEC materials+identification μ | 1.330* | 0.161 | 1.424* | 0.242 | 1.386** | 0.271 |
| | Supplies and commodities+non-pharmaceutical+job aids/IEC materials+identification σ | −0.562* | 0.283 | −0.060 | 0.140 | 0.924* | 0.359 |
| | Supplies and commodities+non-pharmaceutical+job aids/IEC materials)+Identification + Safety gears μ | 2.281** | 0.225 | 2.746** | 0.346 | 1.836** | 0.306 |
| | Supplies and commodities+non-pharmaceutical+job aids/IEC materials)+Identification + Safety gears σ | 1.149** | 0.255 | 1.160** | 0.346 | 1.113* | 0.516 |
| IGA | KES 100 000 | Ref. (0) | | | | | |
| | KES 150 000 μ | −0.119 | 0.107 | −0.109 | 0.170 | −0.146 | 0.147 |
| | KES 150 000 σ | 0.114 | 0.106 | 0.184 | 0.333 | 0.141 | 0.106 |
| Monthly stipend (thousands of KES) | Monthly stipend μ | 0.543** | 0.269 | 0.470** | 0.175 | 0.697** | 0.175 |
| | Monthly stipend σ | 0.766** | 0.153 | 0.865 | 0.984 | 0.753** | 0.236 |
| Decision makers | | 211 | | 109 | | 102 | |
| Observations | | 5064 | | 2616 | | 2448 | |
| Log likelihood | | −1088.899 | | −554.010 | | −521.411 | |
| McFadden's R$^2$ | | 0.104 | | 0.129 | | 0.085 | |

Continued

**Table 4** Continued

| Attribute | Attribute level | Whole sample | | Kilifi | | Bungoma | |
|---|---|---|---|---|---|---|---|
| | | β | SE | β | SE | β | SE |
| Akaike's information criterion | | 2213.798 | | 1144.02 | | 1078.822 | |
| Bayesian information criterion | | 2331.336 | | 1249.669 | | 1183.276 | |

All attributes were random and normally distributed except monthly stipend which was restricted to a lognormal distribution. Values of monthly stipend are in thousands of KES. Statistical significance: \*\*at 0.01 level, \*at 0.05 level. β is the coefficient, SE is the robust SE, μ is the mean, σ is the SD and ASC is alternative specific constant for alternative. All models used 1000 Halton draws

ASC, alternative specific constant; CHEW, community health extension worker; CHV, community health volunteer; CU, community unit; IEC, information, education and communication; IGA, income-generating activities.

recognition at the community level if everything else was kept constant. Additionally, CHVs were willing to accept a KES 4805 (US$48.05) increase in monthly stipend for an incentive package that provided motorcycles to one that provided bicycles only. The magnitude of the increase was greater among CHVs in Kilifi county KES 5917 (US$59.17) than Bungoma KES 3913 (US$39.13).

Furthermore, CHVs in the whole sample were willing to accept a KES 2514 (US$25.14) reduction in monthly stipend for an incentive package that additionally provided identification and a KES 4281 (US $42.81) decrease for an incentive package that additionally provided identification and safety gears to one that only provided supplies and commodities, non-pharmaceuticals and job aids when everything else was kept constant. These reductions in monthly stipends were also greater in Kilifi county than Bungoma. Overall, there was significant heterogeneity in WTA estimates across respondents in most attributes as denoted by the statistically significant SD.

### Interactions

Table 6 shows the model that present results of the interactions between incentive attributes and CHVs' characteristics. Results show that younger CHVs significantly preferred higher salaries when everything else was kept constant. The interaction between monthly stipend and gender suggested that male CHVs preferred higher salaries than female CHVs if everything else is kept constant. However, this interaction was not statistically significant. Overall, both interactions revealed significant heterogeneity in preferences across respondents.

### DISCUSSION

Our study shows that transport was considered the most important attribute, followed by tools of trade and monthly stipend. Some variations in these preferences were observed between Kilifi and Bungoma counties perhaps due to rurality and varying socioeconomic contexts. Critical variations in preferences were observed based on CHV's age and gender. Our result shows that older CHVs significantly preferred incentives with lower

salaries when everything else was kept constant. Perhaps because older CHV may have experienced voluntariness for long and are therefore content with lower salaries. Younger male CHVs had a stronger preference for higher salaries, compared with older female CHVs, an observation associated with gender roles, perceptions of workload, resources and logistics, which are barriers to CHW performance.[40] It is plausible that gender roles affect salary preference and may have implications on retention, a policy issue that any incentive structure need to consider.[41] Workload on the other hand was discussed in the context of use of paper tools or CHWs covering large areas with expectations of reporting household visits monthly regardless of spread, limiting their ability to engage in personal activities and responsibilities.[42] These results contrast with studies from India, which showed that age did not affect CHW salary preferences.[43] Overall, heterogeneity in preferences across respondents is a feature that has been reported elsewhere, with preferences being influenced by education level, having another paid job or being the main household earner.[43]

CHVs had a negative preference for incentive that provided recognition at the facility and award mechanisms for the best performing CHVs compared with jobs that provided recognition at the community level. These results are augmented by WTA estimates, which suggest that CHVs were willing to accept a KES 1060 (US $10.60) increase in monthly stipend for a job that awarded the best performing CHV to one that provided recognition at the community level if everything else was kept constant. Recognition was discussed as a key motivator that will enhance their work. For example, recognition at community level could be as simple as being given a chance to speak at the community public events or benchmarking to other sites to share learnings. Since CHWs are often believed to be vehicles for facilitating community agency and triggering social change,[44] recognition is central to maintaining trust between communities and CHW. Community recognition has been shown to influence their status in their locality, which might confer other benefits beyond their role as CHVs as has been reported

**Table 5** Main effects mixed multinomial logit model in WTP space

| Attribute | Attribute level | Whole sample | | Kilifi | | Bungoma | |
|---|---|---|---|---|---|---|---|
| | | β | SE | β | SE | β | SE |
| ASC | ASC | Ref. (0) | | | | | |
| | ASC μ | 0.015 | 0.405 | 0.905 | 1.118 | −0.461 | 0.255 |
| | ASC σ | 0.516 | 0.268 | −0.293 | 2.632 | −0.355 | 0.194 |
| Recognition | Recognition at community level | Ref. (0) | | | | | |
| | Recognition at facility μ | 0.025 | 0.398 | −1.004 | 1.056 | 0.746 | 0.418 |
| | Recognition at facility σ | 2.821** | 0.771 | 3.672* | 1.446 | −2.250** | 0.358 |
| | Award mechanism for best performing CHV μ | 1.060** | 0.375 | 1.169 | 0.841 | 0.769* | 0.328 |
| | Award mechanism for best performing CHV σ | 2.999** | 0.636 | 4.261* | 1.648 | −1.816** | 0.182 |
| Transport | Bicycles for CHV | Ref. (0) | | | | | |
| | Motorcycle for CHV/CU μ | 4.805* | 0.819 | 5.917* | 2.481 | 3.913** | 0.338 |
| | Motorcycle for CHV/CU σ | 4.271** | 0.972 | 6.047** | 1.916 | 2.524** | 0.337 |
| | Motorcycle for CHEW and bicycle for CHV μ | −0.183 | 0.677 | 1.755 | 1.984 | −1.317** | 0.381 |
| | Motorcycle for CHEW and bicycle for CHV σ | 0.776 | 0.630 | −0.859 | 1.290 | 1.188** | 0.185 |
| Tools of trade | Supplies and commodities+non-pharmaceutical+job aids/IEC materials | Ref. (0) | | | | | |
| | Supplies and commodities+non-pharmaceutical+job aids/IEC materials+identification μ | −2.514** | 0.657 | −3.551** | 1.341 | −1.504** | 0.315 |
| | Supplies and commodities+non-pharmaceutical+job aids/IEC materials+identification σ | −0.317 | 0.483 | −0.098 | 0.465 | −0.563* | 0.284 |
| | Supplies and commodities+non-pharmaceutical+job aids/IEC materials+identification+safety gears μ | −4.281** | 1.131 | −6.935** | 2.428 | −2.173** | 0.348 |
| | Supplies and commodities+non-pharmaceutical+job aids/IEC materials+identification+safety gears σ | 2.009** | 0.640 | 2.824 | 1.473 | 1.297** | 0.209 |
| IGA | KES 100 000 | Ref. (0) | | | | | |
| | KES 150 000 μ | 0.190 | 0.224 | 0.164 | 0.607 | 0.217 | 0.148 |
| | KES 150 000 σ | 0.159 | 0.353 | −0.486 | 1.937 | 0.025 | 0.136 |
| Monthly stipend (thousands of KES) | Monthly stipend μ | 0.715* | 0.302 | 0.449 | 0.399 | 2.001 | 1.342 |
| | Monthly stipend σ | 0.524 | 0.439 | 0.212 | 0.675 | 2.916 | 3.664 |
| Decision makers | | 211 | | 109 | | 102 | |
| Observations | | 5064 | | 2616 | | 2448 | |
| Log likelihood | | −1099.937 | | −559.880 | | −525.174 | |
| Akaike's information criterion | | 2235.874 | | 1155.759 | | 1086.348 | |
| Bayesian information criterion | | 2353.412 | | 1261.409 | | 1190.803 | |

All attributes were random and normally distributed except monthly stipend which was restricted to a lognormal distribution. WTP values and monthly stipend are in thousands of KES. Statistical significance: **at 0.01 level, *at 0.05 level. β is the coefficient, SE is the robust SE, μ is the mean, σ is the SD and ASC is alternative specific constant for alternative. All models used 1000 Halton draws

ASC, alternative specific constant; CHEW, community health extension worker; CHV, community health volunteer; CU, community unit; IEC, information, education and communication; IGA, income-generating activities; WTP, willingness to pay.

in Uganda, with an enhanced status being conferred to them that derive pride and greater access to help or information and being consulted on a range of problems.[29]

However, we observed variations in role of recognition with geographical location. For example, in Kilifi county, a coastal mixed urban and rural settings unlike Bungoma

**Table 6** Mixed multinomial logit model with interactions (gender and age of the CHV): preference estimates

| Attribute | Attribute level | β | SE |
|---|---|---|---|
| ASC | ASC | Ref. (0) | |
| | ASC μ | 0.127 | 0.215 |
| | ASC σ | −0.076 | 0.239 |
| Recognition | Recognition at community level | Ref. (0) | |
| | Recognition at facility μ | −0.149 | 0.202 |
| | Recognition at facility σ | 1.675** | 0.296 |
| | Award mechanism for best performing CHV μ | −0.781** | 0.216 |
| | Award mechanism for best performing CHV σ | 1.696** | 0.267 |
| Transport | Bicycles for CHV | Ref. (0) | |
| | Motorcycle for CHV/CU μ | −2.618** | 0.331 |
| | Motorcycle for CHV/CU σ | 2.334** | 0.347 |
| | Motorcycle for CHEW and bicycle for CHV μ | 0.395 | 0.348 |
| | Motorcycle for CHEW and bicycle for CHV σ | 0.817** | 0.311 |
| Tools of trade | Supplies and commodities+non-pharmaceutical+job aids/IEC materials | Ref. (0) | |
| | Supplies and commodities+non-pharmaceutical+job aids/IEC materials+identification μ | 1.536** | 0.216 |
| | Supplies and commodities+non-pharmaceutical+job aids/IEC materials+identification σ | −0.653* | 0.323 |
| | Supplies and commodities+non-pharmaceutical+job aids/IEC materials+identification+safety gears μ | 2.525** | 0.306 |
| | Supplies and commodities+non-pharmaceutical+job aids/IEC materials+identification+safety gears σ | 1.206** | 0.271 |
| IGA | KES 100 000 | Ref. (0) | |
| | KES 150 000 μ | −0.592 | 0.587 |
| | KES 150 000 σ | 0.090 | 0.322 |
| Monthly stipend (thousands of KES) | Monthly stipend μ | 1.182** | 0.290 |
| | Monthly stipend σ | 0.561** | 0.109 |
| Monthly stipend ##women | Monthly stipend ##women μ | −0.214 | 0.121 |
| | Monthly stipend ##women σ | 0.212* | 0.086 |
| Monthly stipend ##age | Monthly stipend ##age μ | −0.011* | 0.005 |
| | Monthly stipend ##age σ | 0.006* | 0.003 |
| IGA ##women | IGA (KES 150,000) ##women μ | 0.133 | 0.224 |
| | IGA (KES 150,000) ##women σ | −0.085 | 0.154 |
| IGA ##age | IGA (KES 150,000) ##age μ | 0.009 | 0.012 |
| | IGA (KES 150,000) ##age σ | 0.001 | 0.018 |
| Decision makers | | 211 | |
| Observations | | 5064 | |
| Log likelihood | | −1081.439 | |
| McFadden's R$^2$ | | 0.107 | |
| Akaike's information criterion | | 2214.877 | |
| Bayesian information criterion | | 2384.655 | |

## denotes interactions, age is in years, monthly stipend is in thousands of KES. All attributes were random and normally distributed except monthly stipend and its interactions with gender and with age which were restricted to a lognormal distribution. Statistical significance: **at 0.01 level, *at 0.05 level. β is the coefficient, SE is the robust SE, μ is the mean, σ is the SD and ASC is alternative specific constant for alternative. All models used 1000 Halton draws.

ASC, alternative specific constant; CHEW, community health extension worker; CHV, community health volunteer; CU, community unit; IEC, information, education and communication; IGA, income-generating activities.

an agrarian setting, CHVs preferred jobs that provided recognition at the facility to recognition at community. These differences might be due to cultural orientation or benefits accrued from serving in the facility that might elevate CHVs to a same status as health facility-based providers.

Lack of transport for CHWs is reported as a factor limiting their work performance.[29 40] Our results show that CHVs had a negative preference for jobs that provided motorcycles to be used by the CHU and positive preference for jobs that provided motorcycles to the CHU and bicycles to the CHV compared with those that provided bicycles only. These results illustrate the value of transport that facilitates CHV to conduct routine visits. Preferences for bicycles for themselves and a motorcycle in every CHU is probably based on the ability to improve community facility linkages by transportation of clients to the hospital. Community referrals to a facility is a major barrier to optimal service use where clients who do not have means of transport fail to comply with referrals leading to poor outcomes. Motorcycles can also be used communally as a source of income when not busy serving patients. However, issues of maintenance cost and managing communally owned motorbikes may be an avenue for disagreement and might be a complex policy option to implement. These results echo experiences from Uganda, which showed the CHW prefer programmes with transport in the form of bicycles, which would facilitate visiting clients or going to the health centre for supervisory meetings or access supplies, but if non-existent conferred additional financial cost.[29]

CHVs significantly preferred jobs that provided more tools of trade to those that provided limited tools with significant heterogeneity in preference across respondents. Having reporting tools and job aids replenished timely facilitate accuracy in reporting as well, orientation to new updates and facilitate CHW performance. Provision of identification materials, such as T-shirt, badges, caps, branded bags and reflector jackets were described as useful, especially during the rainy seasons to maintain the training materials. In addition, provision of essential requirements in the form of CHV kit, including pharmaceuticals and non-pharmaceuticals for work, was considered important. This may mean that policies that provide CHVs with essential tools of trade is likely to influence their performance, but will have to be structured in a way that ensures limited stockouts to facilitate continuity of service to communities. In summary, other studies have shown that transport constraints and lack of supplies hinder CHW performance,[29 45] thus a useful non-financial incentive.

Finally, relative importance estimates indicate that transport was considered the most important attribute followed by tools of trade and monthly stipend. The least important attribute was IGA. These findings have significant policy direction in the sense that an incentive structure which combines effective transport system for CHVs, essential tools of trade to enable them to perform their roles and monthly stipend will be critical for the success of CHV programme. However, the levels may need to be contextualised to ensure that the implementation is feasible and sustainable. It might also be important to consider geographical variations even within a country. For example, relative importance varied with Kilifi county considering tools of trade as the most important attribute followed by transport and monthly stipend while Bungoma CHVs rated transport as the most important attribute followed by monthly stipend and then tools of trade. These findings support the growing evidence of the importance of the non-financial interventions in motivating CHWs in developing countries.[43]

DCE is a stated preference approach that enabled us to study incentive packages that did not exist. We were able to quantify CHVs preferences for the attributes of incentive alternatives, relative importance they placed on the attributes, and quantify trade-offs of incentive attributes CHVs were willing to make. This provides useful information for policymakers on the attributes of a good incentive package for CHVs in Kenya and other similar contexts. However, since DCE is a stated preference approach, they are prone to bias as they present hypothetical alternatives to respondents. Furthermore, our DCE adopted a forced-choice elicitation format (lack of opt-out), which might have exacerbated hypothetical bias and affected our WTA estimates.[46] Nonetheless, we used qualitative methods to identify and validate our attributes and levels with potential study participants and policymakers, which might have reduced hypothetical bias. A potential value of the DCE beyond the menu of incentives options that CHW prefer may be to practically test the results by implementing the preferred incentive attributes and assessing how it influences the motivation and their retention. Although our results show heterogeneity in terms of incentive preferences for CHVs in different locations, the strength is that it illustrates the value of diverse settings and how that can be used to ensure context-specific adaptation of the menu of incentives that a country can adopt to make use of limited resources.

## CONCLUSION

Our study confirms that a bundled incentive of both financial and non-financial packages is necessary to provide a conducive working environment for CHVs. The menu of options relevant for CHVs in Kenya include transport that was considered the most important attribute followed by tools of trade and monthly stipend. The least important attribute was IGA. Policy decisions should be contextualised to include these attributes to facilitate CHW satisfaction and performance.

**Contributors** TA co-conceptualised the study, led the analysis and drafted the manuscript. DM supported in data collection, overall study implementation, data management, analysis and reviewed the manuscript. MO supported in technical design of the discrete choice experiment (DCE), guided analysis and reviewed the manuscripts for technical content. CN supported in design of the DCE and reviewed the manuscript. GO supported in technical design of the DCE and reviewed the

manuscripts for technical content. DK and JW supported in design of the DCE and reviewed the manuscripts for technical content. CW and SA are lead investigators in the overall study, co-designed the DCE study, framing of this manuscript, and contributed significantly to writing and reviewing drafts. All contributing authors provided inputs and approved the final draft for submission.

**Funding** This paper is based on research funded by the Bill & Melinda Gates Foundation ID OPP1174594. The findings and conclusions contained within are those of the authors and do not necessarily reflect positions or policies of the Bill & Melinda Gates Foundation.

**Competing interests** None declared.

**Patient consent for publication** Not required.

**Ethics approval** The research protocol was approved by the Population Council's Institutional Review Board (PC IRB 872) and African Medical and Research Foundation Ethical Review Board ESRC P572-2018. All the participants gave written consent for the discrete choice experiment survey.

**Provenance and peer review** Not commissioned; externally peer-reviewed.

**Data availability statement** Data used for this paper will be made available upon reasonable request.

**ORCID iDs**
Timothy Abuya http://orcid.org/0000-0001-8815-8299
George Odwe http://orcid.org/0000-0002-5330-2366

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
