## [Reviewer comments · BMJ Open]

ARTICLE DETAILS

TITLE (PROVISIONAL)	Incentive preferences for community health volunteers in Kenya: Findings from a Discrete Choice Experiment
AUTHORS	Abuya, Timothy; Mwanga, Daniel; Obadha, Melvin; Ndwiga, Charity; Odwe, George; Kavoo, Daniel; Wanyugu, John; Warren, Charlotte; Agarwal, Smisha

VERSION 1 – REVIEW

REVIEWER	Blake Angell The George Institute for Global Health
REVIEW RETURNED	03-Feb-2021

GENERAL COMMENTS	Thanks for the chance to review this manuscript. It explores an interesting and important area, I have a few comments that the authors might like to consider prior to the papers publication.  - Table 1 seems to suggest that all attribute levels are dummy coded. Have the authors considered effects coding? Effects coding will allow the impact of reference groups to be separated from that of the constants, I think most recent studies in the field would have used such coding (I will list a few examples below). How to do so is explained in reference 21 of the manuscript. - Did the authors try modelling the stipend and IGA as categorical variables to test for non-linear effects? Assuming a linear relationship from just two levels in particular (for IGA attribute) may be quite a leap. - It's not entirely clear to me how the WTA estimates quoted in the text emerge from Table 5 and I'm not familiar with this approach used by the authors. It would be helpful to explain in more detail. - Table 4 – what is the coefficient for monthly stipend referring to? The effect per 1000 KES perhaps? This should be made clear - Why are there no pseudo R2 stats for the sub group models in Table 4? - Could the authors provide an example choice set presented to respondents? - The authors have seemingly collected enough data to have a good idea of the baseline situation facing most CHVs for each attribute, it might be useful to use this information and the results of the DCE to generate predictions for job uptake under the baseline situation and different bundles of policy packages (made up of the different attributes). Minor point  - In the introduction, the acronym for community health volunteers is introduced as CHW rather than CHV which is used throughout the rest of the paper Similar studies with effects coding  - Mandeville, Kate L., et al. "The use of specialty training to retain
--

	doctors in Malawi: A discrete choice experiment." Social science & medicine 169 (2016): 109-118. - Umar, Nasir, et al. "Toward improving respectful maternity care: a discrete choice experiment with rural women in northeast Nigeria." BMJ global health 5.3 (2020): e002135. - Quaife, Matthew, et al. "Divergent preferences for HIV prevention: a discrete choice experiment for multipurpose HIV prevention products in South Africa." Medical Decision Making 38.1 (2018): 120-133. - Abdel-All M, Angell B, Jan S, et al What do community health workers want? Findings of a discrete choice experiment among Accredited Social Health Activists (ASHAs) in India BMJ Global Health 2019;4:e001509.
--	--

REVIEWER	Shun-Ping Li Shandong Univ
REVIEW RETURNED	04-Feb-2021

GENERAL COMMENTS	This paper focus on the incentive preferences for community health volunteers in Kenya using a discrete choice experiment. Given the evidences of DCEs used in low income setting for various cadres of health workers and especially among community health workers are limitation, this is an important research. The careful development and piloting of the survey instrument is a positive feature of the paper. The methodology is well established and appropriate to the problem. There are a number of changes that could be made to strengthen the paper further. Some of the language and terminology used is not as clear as it could be.  1. The description of attributes are not explained clearly enough. For example, what do "tools of trade" do that improve motivation, performance, and retention of CHVs? Key concepts need clearer definition. 2. It is better to present a choice set in the experimental design to clearly present how the incentive preferences of community health volunteers are assessed. 3. Some information about data collection is unclear. When is the data collected? How is the data sampled from two sub counties of Kilifi and Bungoma? What is the response rate of this survey? 4. This is a cross-sectional study, why a panel mixed multinomial logit model was used to analyze? 5. The main effects panel mixed multinomial logit model in WTP space only gives the β coefficients, it doesn't show the amount of money intuitively. 6. In the analysis of socio-demographic information, age was treated as an ordered categorical variable. Therefore, chi-square test is not appropriate, it's better to use rank sum test. 7. The authors regarded forced-choice as a disadvantage, so why not have an opt-out choice in the experimental design? 8. The presentation of tables are not standard. It would be better to use three-line tables. 9. The article needs further polishing. Be consistent in tenses(is/was) and abbreviations(DCE/DCEs).
---

VERSION 1 – AUTHOR RESPONSE

Reviewer: 1 Mr. Blake Angell, The George Washington University Milken Institute of Public Health

Comments to the Author:

Thanks for the chance to review this manuscript. It explores an interesting and important area; I have a few comments that the authors might like to consider prior to the paper's publication.

Comments 1: Table 1 seems to suggest that all attribute levels are dummy coded. Have the authors considered effects coding? Effects coding will allow the impact of reference groups to be separated from that of the constants, I think most recent studies in the field would have used such coding (I will list a few examples below). How to do so is explained in reference 21 of the manuscript.

Response 1: We thank the reviewer for this comment. We think that there is no need to do effects coding. Our submission is based on Daly, Dekker, and Hess (2016). The authors argue that dummy and effects coding are theoretically equivalent, they address key concerns. First, Effects and dummy coding give the same model fit to the data though the actual coefficients will differ between the two coding schemes. The authors state that proponents of effects coding argue that in dummy coding, the base levels of the categorical variable confound with base levels of the alternative specific constants. The authors do not deny that this scenario cannot happen but argue that this misses the crucial point in choice modelling that only differences in utility matter when comparing between alternatives and between values for a given attribute.

Daly, Dekker, and Hess (2016) continue to state that "what matters is not the absolute contribution of say β of an attribute level (e.g., level 1 or base level) to the utility, but the change in utility say for moving from one attribute-level to another attribute-level (e.g., level 1, base level), and how that compares to moving from alternative A to alternative B. Differences in utility between individual levels for a given variable are not affected by which normalization is used, and as a result, this confounding does not actually matter." Daly, Dekker, and Hess (2016) (p4).

Furthermore, effects coding is not "reference less" (without a reference point). Parameter estimates are interpreted to a base value. The reference is the unweighted average of the values of all levels. In dummy coding the reference is the level set as the base. Second issue raised by the Daly, Dekker, and Hess (2016) is usefulness of the outputs which centres around interpretation of coefficients. Dummy coding is easier to interpret for readers than effects coding.

For these reasons, dummy coding continues being the most dominant coding scheme across the four major fields that use choice modelling i.e., marketing, transport economics, environmental economics, and health economics. The claims that effects coding offers numerous advantages over dummy coding are usually overstated.

Reference

Daly, A., Dekker, T., and Hess, S. (2016). *Dummy coding vs effects coding for categorical variables: Clarifications and extensions. Journal of Choice Modelling.* 21 (.), p36-41.

Comment 2: Did the authors try modelling the stipend and IGA as categorical variables to test for non-linear effects? Assuming a linear relationship from just two levels (for IGA attribute) may be quite a leap.

Response 2: This is an interesting comment as there is usually a trade-off: to keep a variable as continuous or transform it into a categorical one. Initially, we kept both variables in their continuous form to retain as much information as possible (they are both in monetary units i.e., Kenya shillings). Therefore, a shilling (KES 1) change in stipend or IGA is associated with an 'X' change in utility. But we have followed the reviewer's advice and converted IGA into a categorical variable. All models in the paper now have IGA as a categorical variable. The stipend attribute was retained as a continuous variable to compute meaningful WTP measures.

Comment 3: It's not entirely clear to me how the WTA estimates quoted in the text emerge from Table 5 and I'm not familiar with this approach used by the authors. It would be helpful to explain in more detail.

Response 3: In multinomial and conditional logit models, WTA/WTP estimation is straight forward as all attributes are fixed. We would have taken the ratio between the negative coefficient of the non-monetary attributes and the monetary one. However, if any of the variables is random, then the normal approach will result in not well-behaved distributions (Hensher, Rose, and Greene, 2015). In MMNL, some or all the attributes are random with a particular distribution such as normal or log normal. Therefore, we compute WTA estimates in WTP space as outlined by Train and Weeks (2005), and Hole and Kolstad, (2012). Non-monetary attributes coefficients are multiplied by monetary attribute coefficient and WTP distributions derived directly. The WTP distributions are specified as normal distributions while the monetary attribute (stipend) is specified as log normal. The utility functions were written as follows:

$$U_{njt} = \beta_{4n}(\text{Monthly stipend}_{njt} + \beta_0 + \beta_1 * \text{Recognition at facility}_{njt} + \beta_2 * \text{Award mechanism}_{njt} + \beta_3 * \text{IGA}_{njt} + \beta_5 * \text{Motorcycle}_{njt} + \beta_6 * \text{Motorcycle and Bicycle}_{njt} + \beta_7 * \text{Identification}_{njt} + \beta_8 * \text{Identification and safety gear}_{njt}) + \varepsilon_{njt}$$

We refer the reviewer to the references below above for more details about estimating an MMNL model in WTP space to derive WTP/WTA measures directly.

References

Hensher DA, Rose JM, Greene WH. 2015. *Applied Choice Analysis*, Cambridge: Cambridge University Press.

Hole AR, Kolstad JR. 2012. *Mixed logit estimation of willingness to pay distributions: a comparison of models in preference and WTP space using data from a health-related choice experiment*. *Empirical Economics* 42: 445–69.

Train K, Weeks M. 2005. Discrete choice models in preference space and willingness-to-pay space. In: Scarpa R, Alberini A (eds). Applications of Simulation Methods in Environmental and Resource Economics. Dordrecht: Springer Netherlands

Comment 4: Table 4 – what is the coefficient for monthly stipend referring to? The effect per 1000 KES perhaps? This should be made clear.

Response 4: Yes. We have now updated this in the text. Yes. Monthly stipend is in Thousands of KES. We have updated this in the notes under each table by stating that “monthly stipend is in thousands of KES.”

Comment 5: Why are there no pseudo R2 stats for the subgroup models in Table 4?

Response 5: We have now included the McFadden’s R2 stats in Table 4. We have additionally added the Akaike Information Criteria (AIC) and Bayesian Information Criterion (BIC) stats which are also used to assess model fit.

Comment 6: Could the authors provide an example choice set presented to respondents?

Response 6: Thank you for this comment. We have provided an example of the choice set presented to respondents.

Comment 7: The authors have seemingly collected enough data to have a good idea of the baseline situation facing most CHVs for each attribute, it might be useful to use this information and the results of the DCE to generate predictions for job uptake under the baseline situation and different bundles of policy packages (made up of the different attributes).

Response 7: Thank you for this observation. Yes true we have data to provide some information on the situation facing most CHVs and our plan is to write a related paper on this component. After we publish the DCE results we will be writing another paper on this

Minor point

Comment 8: In the introduction, the acronym for community health volunteers is introduced as CHW rather than CHV which is used throughout the rest of the paper

Response 8: Thank you. This is amended to CHVs

Reviewer: 2
Dr. Shun-Ping Li, Shandong University

Comments to the Author:

This paper focus on the incentive preferences for community health volunteers in Kenya using a discrete choice experiment. Given the evidence of DCEs used in low income setting for various cadres of health workers and especially among community health workers are limitation, this is an important research. The careful development and piloting of the survey instrument is a positive feature of the paper. The methodology is well established and appropriate to the problem. There are several changes that could be made to strengthen the paper further. Some of the language and terminology used is not as clear as it could be.

Comment 1. The description of attributes is not explained clearly enough. For example, what do “tools of trade” do that improve motivation, performance, and retention of CHVs? Key concepts need clearer definition.

Response 1: this has been updated in the table.

Comment 2. It is better to present a choice set in the experimental design to clearly present how the incentive preferences of community health volunteers are assessed.

Response 2: Thank you for this comment. We have provided an example of the choice set presented to respondents.

Comment 3. Some information about data collection is unclear. When is the data collected? How is the data sampled from two sub counties of Kilifi and Bungoma? What is the response rate of this survey?

Response 3: Thank you for this comment. We have provided some more information on data collection period, sampling, and response rates in the text.

Comment 4: This is a cross-sectional study, why a panel mixed multinomial logit model was used to analyze?

Response 4: Though the study was cross-sectional, the DCE dataset is not. It is panel since 1 individual answers multiple choice tasks. In our case, 1 individual answered 12 choice tasks. Therefore, 1 individual will have at least 12 observations. These choice tasks are not independent of each other. A panel mixed logit was used to account for within respondent correlation. i.e., the panel structure of our data as one respondent faces 12 choice sets. The “mixlogit” command on stata is by default a panel mixed logit.

Comment 5. The main effects panel mixed multinomial logit model in WTP space only gives the β coefficients, it doesn't show the amount of money intuitively.

Response 5: The coefficients of the panel mixed logit model in WTP space when multiplied by 1000 show the amount of money in Kenya Shillings. Multiplying by 1000 is because we scaled down the monetary attribute (stipend) by dividing it by 1000 so that the model could converge faster. We did put a note under table 5 stating that the WTA values are in thousands of Kenya Shillings. We have now included a sentence in the main text, results sections, stating that “Table 5 shows WTA values in thousands of Kenya Shillings (KES).”

Comment 6. In the analysis of socio-demographic information, age was treated as an ordered categorical variable. Therefore, chi-square test is not appropriate, it's better to use rank sum test.

Response 6: Thank you for this comment. This variable was collected in continuous form, so we have amended and analyzed using independent sample t-test.

Comment 7. The authors regarded forced-choice as a disadvantage, so why not have an opt-out choice in the experimental design?

Response 7: We wanted CHVs to choose an incentive alternative rather than the opt-out as we wanted the study to reflect conditional demand. CHV are currently volunteers without pay and offering them incentive alternatives with pay in a forced-choice format was the best way to elicit preferences in our case. However, we do understand the implications of removing the opt-out which we did admit in the manuscript that it might exacerbated hypothetical bias.

Comment 8. The presentation of tables are not standard. It would be better to use three-line tables.

Response 8: Thank you for this observation. We have used the three-line tables format.

Comment 9. The article needs further polishing. Be consistent in tenses(is/was) and abbreviations (DCE/DCEs).

Response 9: Thank you. We have amended this and written in full where we are using DCE (without s) for the first time and where we are using DCEs (with an s) for the first time.

VERSION 2 – REVIEW

REVIEWER	Blake Angell The George Institute for Global Health
REVIEW RETURNED	25-May-2021

GENERAL COMMENTS	Thanks to the authors for their detailed response to the previous points raised. I have only a few follow-up points for consideration. - Table 1 appears unfinished (the description column is empty)
--

	- Data analysis section - 'Panel' has been removed from the description of the models used – has the model changed? I think what the authors describe should be labelled as a panel mixed model.
--	---